# Regulation of Osteoclast Differentiation and Skeletal Maintenance by Histone Deacetylases

**DOI:** 10.3390/molecules24071355

**Published:** 2019-04-06

**Authors:** Bora Faulkner, Kristina Astleford, Kim C. Mansky

**Affiliations:** 1Department of Genetics, Cell Biology and Development, University of Minnesota, Minneapolis, MN 55455, USA; kuyom001@umn.edu; 2Diagnostic and Biological Sciences, School of Dentistry, University of Minnesota, Minneapolis, MN 55455, USA; astl0008@umn.edu; 3Department of Developmental and Surgical Sciences, University of Minnesota, Minneapolis, MN 55455, USA

**Keywords:** osteoclasts, HDAC, transcription, gene, acetylation, differentiation and resorption

## Abstract

Bone is a dynamic tissue that must respond to developmental, repair, and remodeling cues in a rapid manner with changes in gene expression. Carefully-coordinated cycles of bone resorption and formation are essential for healthy skeletal growth and maintenance. Osteoclasts are large, multinucleated cells that are responsible for breaking down bone by secreting acids to dissolve the bone mineral and proteolytic enzymes that degrade the bone extracellular matrix. Increased osteoclast activity has a severe impact on skeletal health, and therefore, osteoclasts represent an important therapeutic target in skeletal diseases, such as osteoporosis. Progression from multipotent progenitors into specialized, terminally-differentiated cells involves carefully-regulated patterns of gene expression to control lineage specification and emergence of the cellular phenotype. This process requires coordinated action of transcription factors with co-activators and co-repressors to bring about proper activation and inhibition of gene expression. Histone deacetylases (HDACs) are an important group of transcriptional co-repressors best known for reducing gene expression via removal of acetyl modifications from histones at HDAC target genes. This review will cover the progress that has been made recently to understand the role of HDACs and their targets in regulating osteoclast differentiation and activity and, thus, serve as potential therapeutic target.

## 1. Introduction

A common misconception about the skeleton is that it is static; however, bone is an ever-changing organ that is remodeled through tight coupling of bone resorption followed by formation of new bone [1,2,3]. These processes are performed by bone-resorbing osteoclasts and bone-forming osteoblasts. Skeletal homeostasis depends on strict control over the number of active osteoclasts at any site [4]. Since osteoclasts are a perpetrator of many skeletal diseases, understanding the mechanisms that regulate their activity during the bone remodeling process is necessary [5,6,7].

The dynamic and responsive nature of bone during the remodeling process requires temporal changes in gene expression within the osteoclast lineage [8]. The combination of transcription factors and co-factors binding to DNA sequences plays an important role in chromatin remodeling and cellular signaling events in regulating osteoclasts, thus impacting bone remodeling. Additionally, regulation of gene expression is controlled, in part, by histone deacetylases (HDACs) [9], which are intracellular enzymes that directly affect chromatin structure, transcription factor activity [10], signaling events, and thus affect the activities required for bone remodeling. Understanding how these molecular and cellular switches work has become an important mechanism(s) to consider in regards to developing targeted bone therapies. HDACs are clinically relevant because many molecules that inhibit their activities are used alone or in combination with other drugs to treat diseases such as osteoporosis and cancer [11].

To develop therapies that halt the progression of skeletal diseases, we need to understand how osteoclasts operate, including the mechanisms that regulate their generation, regulation, and bone-resorbing activity. Therefore, transcription factors, co-factors, transcriptional regulators, and co-regulators must be studied to determine their effects on osteoclastogenesis. Recently, a great deal of progress has been made in investigating these mechanisms and the bone degradation process using experimental mouse models and patients with abnormal bone phenotypes [5,12,13,14].

In this review, we focus on regulation of osteoclasts by transcriptional factors and introduce important findings on how osteoclasts are regulated by HDACs to exert bone-resorbing activity. We have organized this review according to the different categories of regulation. We will begin by discussing osteoclasts and their transcriptional factors followed by the influence of HDACs on osteoclast differentiation and activity.

## 2. Osteoclast Biology

Bone is comprised of multiple cell types, including osteoblasts, which are mesenchymal-derived cells responsible for synthesizing new bone, osteocytes, which are terminally-differentiated osteoblasts, and bone-degrading osteoclasts, which are hematopoietic in origin [15]. Bone homeostasis and normal function result from the tightly-regulated interactions between osteoblasts, osteocytes, and osteoclasts. To maintain this balance in the adult skeleton, osteoblasts and/or osteocytes produce two necessary cytokines, macrophage colony stimulating factor (M-CSF) and receptor activator of NF-κB ligand (RANKL), to promote osteoclast differentiation [16].

Osteoclasts are giant multinucleated cells of hematopoietic origin that degrade the bone matrix. They are formed by fusion of mononuclear precursors of the monocyte/macrophage lineage and are the primary resorptive cells of the skeleton [4,17]. One of the most remarkable features of the osteoclast is its ruffled membrane, which is the resorptive organelle of the cell. When this membrane comes into contact with bone, skeletal matrix degradation follows [18]. Most skeletal disorders manifest when osteoclast activity is dysfunctional [6,7]. Increased bone mass phenotype (osteopetrosis) is associated with low osteoclast activity, while decreased bone mass phenotype (osteoporosis) is associated with increased osteoclast activity.

Progression from multipotent progenitors into specialized, terminally-differentiated cells involves carefully-regulated patterns of gene expression to control lineage specification and emergence of the cellular phenotype. This process requires coordinated action of transcription factors with co-activators and co-repressors to bring about proper activation and inhibition of gene expression. In the following section, we will describe what is known about transcription factors that regulate osteoclast differentiation (Figure 1).

## 3. Transcriptional Regulators of Osteoclast Gene Expression

### 3.1. PU.1

PU.1 is a member of the ETS-domain transcription factor family [19]. It has been shown that mice lacking PU.1 have a deficiency in both macrophage and osteoclast differentiation, and this established PU.1 as one of the earliest markers of the osteoclast lineage [20]. During osteoclast differentiation, the expression level of PU.1 is unchanged or modestly increased [20,21]. In the early stages of commitment to the monocyte/macrophage lineage, PU.1 upregulates the expression of *Csf1r*, the receptor for M-CSF, one of the essential ligands for osteoclast differentiation [22]. Along with other osteoclast transcription factors, PU.1 regulates the expression of *Tnfrsf11a*, the gene for RANK, the receptor for RANKL, the other essential cytokine for osteoclast differentiation [23]. Lastly, PU.1 along with other osteoclast transcription factors, c-Fos, MITF, and NFATc1, regulates multiple genes necessary for osteoclast differentiation including *cathepsin K* (*Ctsk)*, *Acp5*, the gene for TRAP, *Dc-stamp*, and *Oscar* [21,24,25]. 

### 3.2. MITF

The microphthalmia transcription factor (MITF) is a basic helix-loop-helix-leucine zipper transcription factor [26,27,28]. It has been established that MITF plays an essential role in regulating gene expression during osteoclast differentiation. This observation was based on a bone marrow transplantation experiment, which showed that the osteopetrotic condition in *mi*/*mi* mutant mouse could be cured using bone marrow from a wild-type littermate [29]. Additionally, mononuclear osteoclasts from these *mi/mi* mice were unable to fuse, form a ruffled border, or resorb bone [30]. These results not only confirmed osteoclasts’ hematopoietic origin, but also established MITF as a necessary transcription factor during osteoclast differentiation. MITF along with PU.1 was shown to upregulate the expression of *Tnfrsf11a* [31]. Studies on MITF in osteoclasts have also revealed that it can transcriptionally regulate osteoclast genes (*Acp5*, *Oscar*, and *Ctsk*) along with PU.1 [24,32,33].

### 3.3. CEBPα

The CCAAT/enhancer binding protein-α (C/EBPα) is a member of the C/EBPα family of transcription factors, which act as master regulators of various cellular processes [34]. C/EBPα has been shown to be important for differentiation of myeloid progenitors [35]. Recently, C/EBPα was shown to be highly expressed in osteoclasts [36]. Overexpression studies revealed that PU.1 upregulates the expression of C/EBPα, while C/EBPα upregulates the expression of c-Fos and NFATc1 [36,37]. Forced expression studies revealed that C/EBPα can specify mouse bone marrow cells to become osteoclasts, suggesting that C/EBPα can induce osteoclast lineage priming and plays a role in regulating gene expression during the early stages of osteoclast differentiation [36]. In addition, C/EBPα-deficient mice were shown to have severe osteopetrosis due to impaired osteoclast development [38,39]. In a subsequent study, overexpression of C/EBPα was shown to promote osteoclast differentiation and induce expression of osteoclast genes *Nfatc1*, *Ctsk*, and *Acp5* [40]. Besides its role in regulating osteoclast differentiation, functional experiments revealed that C/EBPα plays a role in bone resorption [40]. Together, all these studies suggest that C/EBPα is essential for osteoclast differentiation and activity.

### 3.4. MEF2

The myocyte enhancer factor 2 (MEF2) family of transcription factors plays an important role in many cellular processes including cell differentiation and apoptosis [41,42,43]. These proteins can affect the expression of many genes by binding with other transcription factors [41,42,43]. One of these binding partners is HDACs. Each individual HDAC contains an MEF2-interacting transcriptional repressor homology domain, suggesting that they have the ability to regulate the MEF2 family [41,42,43]. The MEF2 family consists of four members, MEF2A, B, C, and D [41,42,43]. Osteoclasts primarily express MEF2A and D (Mansky, unpublished observation). Osteoclasts null for MEF2A and D express reduced levels of c-Fos and NFATc1, suggesting that MEF2 may regulate the expression of these transcription factors in osteoclasts (Mansky, unpublished observation).

### 3.5. c-FOS

c-Fos is a member of the activator protein-1 (AP-1) family of transcription factors, and its expression is induced early during osteoclast differentiation [44,45]. c-Fos acts as an important switch between osteoclast and macrophages differentiation, and in its absence osteoclasts do not form [44]. This observation was made in *c-Fos* knockout mice, which develop osteopetrosis due to the inability of cells to commit to the osteoclast lineage [44,46]. c-Fos and NF-κB both target and upregulate *Nfatc1* expression in osteoclasts and at least in part explain why *c-Fos* knockout mice do not develop osteoclasts [45].

### 3.6. NF-κB

Nuclear factor kappa-light-chain-enhancer of activated B cells or NF-κB is a pleiotropic transcription factor part of the Rel subfamily of proteins [47]. In osteoclasts, NF-κB regulates formation, function, and survival [47,48,49]. NF-κB is activated downstream of RANK signaling, which ultimately results in the activation of NFATc1 to induce osteoclastogenesis [47,50]. The inhibitory protein IκB localizes NF-κB in the cytoplasm until it is activated by dimerizing with Rel family proteins promoting nuclear translocation to activate transcription [47,48,49]. Loss of the p50 and p52 subunits of NF-κB results in an osteoporotic bone phenotype in mice, most likely due to the inability to activate *Nfatc1* expression [50].

### 3.7. NFATc1

NFAT is a family of transcription factors that regulate the expression of cytokines and other immunoregulatory genes [51]. NFATc1 mediates RANKL-induced osteoclast formation, and its overexpression in *c-Fos*-deficient cells rescues osteoclastogenesis [45,52]. At early stages of osteoclast differentiation, RANKL increases the stability of NFATc1 protein by stimulating calcineurin-mediated dephosphorylation of NFATc1 in the cytosol, causing it to translocate to the nucleus [53]. However, during late-stage osteoclastogenesis, M-CSF downregulates NFATc1 protein levels, and eventually, NFATc1 is degraded through the ubiquitin-proteasome pathway in the cytoplasm [54]. This degradation is mediated by Cbl-b and c-Cbl ubiquitin ligases in a Src-dependent manner [54]. While NFATc1 is termed the “master regulator” and is sufficient for osteoclast differentiation, ChIP experiments demonstrated that NFATc1 regulates osteoclast gene expression in a complex with PU.1, MITF, and c-Fos [55]. There are no studies yet to demonstrate how or if MEF2 and C/EBPα interact with this complex to regulate osteoclast gene expression.

## 4. Histone Deacetylases

Histones are major regulators of gene expression and transcription, and their state of activeness and inactiveness controls this process [56]. Acetylation is an important modification to histone tails that allows for genes to be transcribed by releasing DNA so they can unwind and become accessible for transcription [56]. There are two enzymes that play an important role in this process: histone acetyltransferases (HATs) and histone deacetylases (HDACs) [56]. HATs are enzymes that will acetylate histone tails, resulting in the relaxation of the chromatin and allowing for transcription factors to bind to their promoters [56]. For transcription to be repressed, HDACs are recruited to remove acetyl groups on the tails, allowing chromatin to rewind around histones [56]. Additionally, some HDAC members have the ability to directly interact with transcription factors to repress transcription [9]. Understanding transcriptional regulation by HATs and HDACs in bone is vital in the study of differentiation and disease in which these two enzymes play a role.

### 4.1. HDAC Classes

There are 18 HDACs within the human genome, each of which are distributed into four different classes: class I HDACs (HDACs 1–3,8), class II HDACs (HDACs 4–7,9,10), class III HDACs (Sirtuins 1–7), and class IV HDAC, HDAC11, [9]. They are divided based on their enzymatic activity, location within the cell, and their sequence homology to yeast [9]. The structure and function of HDACs are similar to one another and are also closely related to that of budding yeast; however, each class has its own unique catalytic domain [9]. Class I HDACs are homologs to yeast RPD3 and are primary localized in the nucleus [9]. They contain a catalytic domain unique to the class I HDACs and predominantly target histone substrates [9]. Class II HDACs are homologs to yeast HDA1 [57,58]. These HDACs are separated into two subclasses; class IIA and class IIB [56]. Class IIA HDACs are transcriptional repressors that are shuttled between the nucleus and the cytoplasm by the chaperone protein 14-3-3 [56,58]. Class IIB HDACs differ from IIA, in that they contain two catalytic domains and are primarily found in the cytoplasm [57]. Class III HDACs, or sirtuins, are homologs to yeast Sir1 [59]. These HDACs are dissimilar to the class I and class II HDACs in that they are NAD^+^ dependent and contain an NAD^+^ binding site [59]. Class III HDACs are known to deacetylate histones and regulate transcription factors [59]. They can regulate transcription factors by altering their binding to DNA, modifying their location within the cell, or changing their interaction with other proteins [59]. HDAC11 is the sole member of the class IV HDACs [60]. It shares a similar structure to that of the class I HDACs; however, little is known about the mechanism by which HDAC11 regulates transcription [60].

### 4.2. HDACs in Bone Development

Bone development occurs throughout life. The cells that are involved in development, growth, and remodeling are chondrocytes, osteoblasts, osteocytes, and osteoclasts. The two main types of bone formation/ossification are intramembranous and endochondral. Endochondral ossification is the process by which long bones and vertebrae form, while intramembranous ossification results in the formation of the flat bones of the skull, mandible, and clavicle. It has been reported that some HDACs have roles in the ossification process. Intramembranous bone defects in mouse models associated with altered class I HDAC expression have been described [61,62,63]. HDACs have also been shown to contribute to different steps of endochondral bone ossification. Germline deletion of HDAC1, 3, and 7 is embryonic lethal, and they die before endochondral ossification begins [64,65,66]. Deletion of HDAC 2, 4, 5, 6, 8, and 9 is not embryonic lethal and does not appear to disrupt early endochondral ossification [67,68,69]. HDAC3 and HDAC4 have been shown to play a role in chondrocyte maturation [65,67,70]. There are currently no published studies demonstrating that HDACs play a role in regulating osteoclast activity during bone development. For a more complete review of HDAC-mediated control of endochondral and intramembranous ossification, see [71].

### 4.3. HDACs and Skeletal Maintenance

Skeletal maintenance requires a tightly-coordinated activity of osteoclast, osteoblasts, and osteocytes [3]. Some of the 18 human HDACs have been shown to play a part in skeletal bone maintenance. Multiple HDACs (1, 3, 4, 5, 6, and 7) are expressed in osteoblasts and have been shown to regulate osteoblast differentiation and gene expression [72,73,74,75,76,77,78], as well as osteocyte activity [79,80]. More complete reviews on the regulation of osteoblasts and osteocyte activity by HDACs are [63,71,81,82]. In Section 5 of this review, we will discuss what is known concerning the role of regulating osteoclasts during the maintenance or remodeling phase of the skeleton.

### 4.4. HDACs and Skeletal Diseases

Currently in humans, mutations in HDAC2, 4, 5, 6, and 8 have been shown to affect the skeleton [83,84,85,86,87,88,89,90,91]. These HDACs mutations results in either loss of function, deletion, or gain of function in patients. Some of the defects in bone caused by these mutations are low bone mineral density, brachydactyly, and skeletal abnormalities in the craniofacial region, the spine, and the growth plates [63]. Further investigations are needed in order to understand the altered HDACs functions. It has also been reported that many HDACs are expressed in human articular cartilage, and the expression of some of the HDACs was even higher in joints from osteoarthritic patients [92,93,94,95,96]. The causative role for the altered HDACs function needs to be investigated. Usually, the amount of bone resorbed by osteoclasts is balanced by the amount of bone formed by osteoblasts. Nevertheless, in increased bone resorption conditions, such as osteoporosis and Paget’s disease, bone resorption surpasses bone deposition, which results in bone loss. The roles of HDACs in regulating osteoclasts in these bone resorption conditions is not understood and is an emerging area of scientific interest.

## 5. Role of HDACs in Osteoclasts

### 5.1. Class I HDACs 

Class I HDACs are thought to be expressed in most cell types [9]. HDAC1 and 2 expression is almost exclusively nuclear, as both proteins lack a nuclear export signal [9]. However, HDAC3 has both a nuclear import and export signal, but is almost always found in the nucleus, perhaps due to the fact that it is recruited to the nucleus via its interactions with HDAC 4, 5, and 7 when they are bound to DNA [97,98,99]. There are currently no published animal model studies analyzing the role of HDAC1, 2, 3, or 8 in osteoclasts using conditional mouse models. These studies are critical to understanding the role of class I HDACs in osteoclast differentiation, as well as skeletal maintenance. Chemical inhibitors and siRNA studies are informative, as they can be used to distinguish between the effects of HDACs regulating osteoclast differentiation versus activity, but they can have off-target effects, making the results difficult to interpret.

#### 5.1.1. HDAC1

HDAC1 is a transcriptional repressor that is present early during osteoclast differentiation [100]. It is expressed in osteoclast precursors and then drops significantly after RANKL stimulation [100]. The main role of HDAC1 in osteoclasts is to act as a co-repressor [101]. It does this by being recruited to the promotors of osteoclast genes such as *Nfatc1* and *Oscar* to prevent their expression [101]. By ChIP analysis, MITF, and PU.1 have been shown to be recruited to *Ctsk* and *Acp5* promoters with M-CSF stimulation along with co-repressors CtBP, Sin3A, and HDAC1 [102].

#### 5.1.2. HDAC2

In contrast to HDAC1, HDAC2 expression increases during osteoclast differentiation [103]. Studies have shown that knock down of HDAC2 in osteoclasts not only inhibits osteoclast differentiation, but also hinders actin ring formation, fusion, and osteoclast activity [103]. HDAC2 was shown to activate Akt, which removed an inhibitor, FoxO1, and demonstrated that HDAC2 is not an inhibitor of osteoclast differentiation.

#### 5.1.3. HDAC3 

HDAC3 is expressed in osteoclast precursors, and its expression remains low during osteoclast differentiation [100]. Knock down of HDAC3 results in inhibition of osteoclastogenesis resulting from the downregulation of osteoclast genes *Nfatc1*, *Ctsk*, and *Dc-stamp* [104]. Loss of HDAC3 expression is similar to the phenotype seen with osteoclasts treated with broad-spectrum HDAC inhibitors (HDIs) such as trichostatin A (TSA) and sodium butyrate (NaB) [105,106].

#### 5.1.4. HDAC8

HDAC8 expression is low during early osteoclast differentiation, and its expression increases during late osteoclast differentiation [100]. Not much is known about HDAC8 in osteoclasts, and as stated above, further animal model studies will need to be done to understand its role in regulating differentiation and activity.

### 5.2. Class II HDACs

Based on cellular signals, class II HDACs are able to shuttle between the cytoplasm and the nucleus. At least in muscle cells, HDAC4, 5, and 7 have a very regulated process of shuttling between the nucleus and cytoplasm [9,107,108]; however, the subcellular localization of class II HDACs has not been well studied in osteoclasts. HDAC4, 5, 7, and 9 all have a binding site in their amino terminus for C-terminal binding protein (CtBP), MEF2, and 14-3-3 [9,98,109]. One interesting question that has emerged pertaining to HDACs 4, 5, 7, and 9 given their similarity is why they are not functionally redundant in osteoclasts. To answer this question, studies need to be performed to identify targets and partners of class II HDACs so as to understand the mechanisms by which class II HDACs regulate osteoclast differentiation and activity.

#### 5.2.1. HDAC4

HDAC4 is expressed in osteoclast precursors, and its expression begins to decrease as osteoclasts become mature [100,110]. Knock down of HDAC4 by shRNA increases osteoclasts in size and number [110]. With the increase in osteoclast differentiation, there was also an upregulation of osteoclast genes (*c-Fos*, *Nfatc1*, *Dc-stamp*, and *Ctsk*) and increases in osteoclast activity [110]. To further study the role of HDAC4 in regulating osteoclast differentiation, conditional deletion of *Hdac4* using *Cfms-Cre* was created and demonstrated to have no gross phenotypic changes in bone structure. More focused analysis surprisingly revealed an increase in bone volume in trabecular bone at three months of age, suggesting an increase in bone mass phenotype (Mansky, unpublished observation). Moreover, bone resorption was reduced in HDAC4cKO mice (Mansky, unpublished observation). The mechanism(s) by which HDAC4 regulates osteoclast differentiation, i.e., targets and partners, is currently unknown and is being evaluated using an osteoclast-specific HDAC4 mouse model (Mansky, unpublished observation).

#### 5.2.2. HDAC5

HDAC5 is expressed later in osteoclast differentiation, having the highest expression around the time of osteoclast fusion [100,110]. Similar to HDAC4, HDAC5 knock down by shRNA results in an increase in osteoclast differentiation, an upregulation of osteoclast genes, and an increase in resorption [110]. Global deletion of HDAC5 in mice demonstrated no gross defect at birth, and mice were of normal size [69]. In two other separate studies, HDAC5 knockout mice were reported to have reduced trabecular bone density at 2–3 months of age [76,79]. One study suggested that loss of HDAC5 expression leads to an increase in RANKL expression by osteoblasts, which stimulated greater bone resorption by the osteoclasts [76]. However, since multiple types of bone cells are being affected in the global HDAC5 null mouse and the mechanism(s) for bone loss is not clear, more focused tissue-specific studies need to be conducted. These mechanisms will be explored using an osteoclast-specific HDAC5 mouse model (Mansky, unpublished observation). Mechanistically, HDAC5 has been shown to deacetylate NFATc1, leading to NFATc1′s instability and is a potential mechanism by which HDAC5 inhibits osteoclast differentiation [111]. 

#### 5.2.3. HDAC6

During osteoclast differentiation, HDAC6, a class IIb HDAC protein, expression was shown to peak around osteoclast fusion and to be expressed predominately in cytoplasm [100,110]. By using shRNA, knockdown of HDAC6 expression in mouse bone marrow macrophages (BMMs) did not display any phenotype, nor did it seem to affect expression of major osteoclast genes [110]. HDAC6 plays an important role in destabilizing the osteoclast cytoskeleton, and inhibiting osteoclast migration and podosome formation [112,113]. This observation revealed a molecular mechanism in which RhoA-mDia2-HDAC6 forms a complex and regulates podosome patterning [112]. It was demonstrated using microinjection experiments that injecting either activated RhoA or mDia2 caused microtubule deacetylation together with podosome belt disruptions [112].

#### 5.2.4. HDAC7

HDAC7 is expressed early in osteoclast differentiation and continues at a low level of expression throughout differentiation [100,110]. HDAC7 has been shown to have a role in regulating osteoclast differentiation [104,114,115]. Osteoclasts with reduced HDAC7 expression resulted in accelerated osteoclast differentiation and increased size of TRAP-positive multinucleated osteoclasts [104,114,115]. This discovery was ascribed to the capacity of HDAC7 to suppress MITF transcriptional activity [115]. It was reported that the N-terminus tail of HDAC7 was sufficient to bind and suppress MITF activity, suggesting that the deacetylase domain of HDAC7 is not required for its repression [115]. Conditional knockdown of HDAC7 using *LysM-Cre*, which targets monocytes and myeloid lineage cells, which includes osteoclasts, was found to enhance osteoclast differentiation and resulted in an osteopenic skeletal phenotype [114,115]. Besides MITF, HDAC7 was shown to inhibit β-catenin activity and cyclin D1 expression in the presence of RANKL [114]. These studies suggest that HDAC7 is a negative regulator of osteoclastogenesis.

#### 5.2.5. HDAC9

HDAC9 is expressed immediately after RANKL stimulation, and then, expression reduces to undetectable levels [100,110]. Osteoclast differentiation and bone resorption were highly elevated in HDAC9 KO mice [100]. Bone marrow transplantation experiments revealed that the osteoclasts defects in HDAC9 KO are intrinsic primarily to the hematopoietic cell lineage, since when HDAC9 KO bone marrow was transplanted to wild-type mice, it resulted in osteopenia, and when wild-type bone marrow was transplanted to HDAC9 KO mice, a rescue occurred [100]. HDAC9 was found to take part in a negative regulatory circuit with PPARγ and RANKL signaling. Both PPARγ and RANKL can inhibit *Hdac9* mRNA expression levels, while HDAC9 forms a complex with NCoR and SMART to inhibit PPARγ activity [100]. In conclusion, these studies demonstrate that HDAC9 expression inhibits osteoclastogenesis. Similar to HDAC5, creating a conditional mouse model of HDAC9 would allow for studies to be performed that analyze the role of HDAC9 in osteoclasts without the confusion of HDAC9′s role in regulating the other cells of the skeleton.

#### 5.2.6. HDAC10

Like HDAC6, HDAC10 is a part of the class IIb HDAC family [116]. The biological role of HDAC10 in osteoclasts is largely unknown; however, what is known is that HDAC10 expression increases during differentiation and peaks around fusion [100,110]. Additionally, suppression of HDAC10 by shRNAs results in increased osteoclastogenesis, expression of osteoclast genes (*c-Fos*, *Nfatc1*, *Dc-stamp*, and *Ctsk*), and osteoclast activity [110]. This suggests that HDAC10 acts as an inhibitor of osteoclast differentiation; however, studies should be done in an animal model to confirm cell culture studies, as well as identify osteoclast targets and partners of HDAC10.

### 5.3. Class III HDACs

Class III HDACs are a family of NAD+-dependent deacetylases known as sirtuins [117]. These proteins are involved in many physiological processes such as cellular metabolism, DNA repair, cell growth, and autophagy [117]. There are seven known sirtuins; however, only three have been studied in osteoclasts. There is not much known about sirtuins 2, 4, 5, and 7 in osteoclasts, and further studies will need to be performed to characterize their roles in regulating osteoclast differentiation and activity.

#### 5.3.1. Sirtuin 1

Sirtuin 1 (SIRT1) is a repressor of osteoclast differentiation and activity by inhibiting RANKL signaling [118]. SIRT1 is able to do this by deacetylating and activating a group of inhibitors of osteoclastogenesis known as forkhead box proteins (FOX) [118]. Loss of SIRT1 expression results in an increase in osteoclast formation and activity due to the increase of acetylated FOX proteins [118].

#### 5.3.2. Sirtuin 3

In osteoclasts, Sirtuin 3 (SIRT3) expression is induced by RANKL stimulation [119]. Loss of SIRT3 in osteoclasts results in osteopenia in mice due to an increase in osteoclast number [119]. This indicates that SIRT3 is a negative regulator of osteoclast differentiation [119]. Additionally, the loss of SIRT3 results in an increase in osteoclast-specific genes such as *Oscar*, *Nfatc1*, and *Atp6v0d2* [119]. Interestingly, SIRT3 does not regulate osteoclast activity, meaning its ability to resorb mineral is unaffected by loss of the gene [119]. Lastly, it is suggested that SIRT3 can negatively regulate osteoclast differentiation by controlling AMPK activity [119].

#### 5.3.3. Sirtuin 6

Sirtuin 6 (SIRT6) is expressed in osteoclasts once monocytes are stimulated with M-CSF and RANKL [120]. SIRT6 acts as a transcriptional repressor through inhibiting NF-κB transcription [120]. This inhibition activity results from the deacetylation of H3K9 on the promotors of target genes for NF-κB [120]. Therefore, overexpression of SIRT6 results in inhibition of osteoclastogenesis, and enhanced osteoclast differentiation is measured when SIRT6 is lost [120]. 

### 5.4. Class IV HDACs

HDAC11, the sole member of class IV HDAC, is expressed late in osteoclast differentiation, where it is most highly expressed after three days of RANKL stimulation [100,110]. HDAC11 appears to be most closely related to HDAC3 and 8; however, its classification has not been determined since its sequence identity with other HDACs is limited [60,121]. Loss of HDAC11 expression in osteoclasts results in an increase in osteoclast differentiation, as well as osteoclast activity. Interestingly, HDAC11 does not affect *c-Fos* or *Nfatc1*, but increases the expression of both *Dc-stamp* and *Ctsk*, suggesting that it plays a role in osteoclast fusion, as well as activity, but independent of regulating *Nfatc1* expression or activity [110]. These results suggest that HDAC11 acts as a repressor; however, further animal model studies will need to be performed to characterize its role in regulating osteoclast differentiation and activity.

## 6. Effects of HDAC Inhibitors on the Skeleton

To date, the majority of studies have analyzed the effects of HDACs and HDIs on the effect of individual bone cells [81,122]. It has become apparent that maintaining a healthy skeleton requires continued crosstalk between osteoclasts, osteoblasts, and osteocytes (Figure 2). Studies assessing the crosstalk between the various cells will need to be done to assess the efficacy of using broad or specific HDIs in treating skeletal diseases.

Since the 1960s, valproate, an HDI, has been used as treatment for epilepsy, bipolar, and other mood disorders. In several patient groups, prolonged exposure to valproate led to a decrease in bone mineral density and an increase in fracture risk [123,124,125,126]. In the valproate studies, the mechanism(s) resulting in the bone loss were not clear, as changes in bone biomarkers were not conclusive [127,128,129,130,131]. Additionally, valproate has other activities besides inhibiting HDACs [132,133]. In animal studies of periodontitis, a broad-acting HDI resulted in bone loss and reduced the number of osteoclasts in the gingiva and alveolar bone [134]. In animal models of rheumatoid arthritis, topical treatment with either broad-spectrum HDI, phenyl butyrate, or TSA suppressed joint swelling and levels of TNF-α with no evidence of joint destruction [135]. Class- and isozyme-specific HDI, NW-21 (targets HDACs1 and 2), MS-275 (targets HDAC1), and BML-275 (targets HDAC6) have been shown to have anti-arthritic activities in rodent models [136,137]. A number of broad-acting and a HDAC6-specific HDI are currently being used in the clinic to treat multiple myeloma [138,139,140,141]. While HDIs may help patients that suffer from skeletal diseases such as osteopetrosis, they may have off-target effects that cause bone loss in other patients. Additionally, these HDIs are broad acting and do not target specific HDACs. Producing new HDIs in the future that target specific HDACs may be more beneficial since HDACs have differing roles in osteoclasts.

### HDAC Inhibitors and Fracture Healing

After initial trauma, bone heals by direct intramembranous or indirect fracture healing involving intramembranous and endochondral bone formation [142]. The bone healing process involves an acute inflammatory response and the recruitment of mesenchymal stem cells so as to initiate the formation of primary cartilaginous callus. The callus is then subjected to revascularization and calcification and eventually remodeled by osteoblasts and osteoclasts [142]. It has been reported that HDIs can increase the bone healing process. Lee et al. [143] reported that they saw greater bone formation in a rabbit calvarial bone fracture model when cyclic depsipeptide largazole was added to a macroporous biphasic calcium phosphate scaffold compared to the scaffold control. The activity of cyclic depsipeptide largazole was attributed to increased expression of Runx2 and BMPs. This finding requires further investigation because, as reported in the previous section, HDI administration in humans has been shown to have negative effects on bone mass. 

## 7. Conclusions

Over the years, we have learned a great deal about the mechanisms by which the osteoclast is formed and resorbs bone. Most of this progress has come about as a result of rapid advances in cell and molecular biology studies. We now know more about the enzymes, signaling pathways, and cytokines that are important for osteoclast formation. While treatments such as HDIs may be beneficial for some skeletal diseases, this new level of understanding will result in the generation of new therapies that specifically target osteoclast generation and/or activity. Despite these advances, several key questions remain unanswered, one of which relates to the detailed mechanism of osteoclast regulation by HDACs, a process that seems to affect its function. More importantly, however, is the issue of the roles of different HDACs at molecular and cellular levels in bone biology since much remains unknown. A better understanding of the roles of HDACs in regulating osteoclasts, especially during bone resorption, will also provide insights into the cellular and molecular mechanisms that regulate skeletal development and homeostasis.

## Figures and Tables

**Figure 1 molecules-24-01355-f001:**
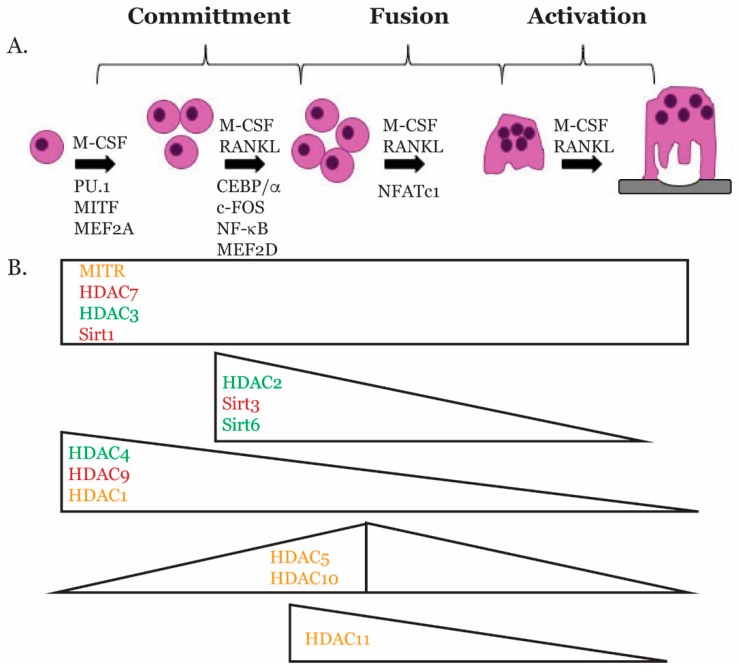
Temporal expression of transcription factors and HDACs during osteoclast differentiation. The cartoon illustrates the expression of transcription factors and HDACs during osteoclast differentiation based on studies presented in this review. HDACs in red inhibit osteoclast differentiation; HDACs in green promote osteoclast differentiation; and the function of HDACs in orange is not known due to the lack of animal studies.

**Figure 2 molecules-24-01355-f002:**
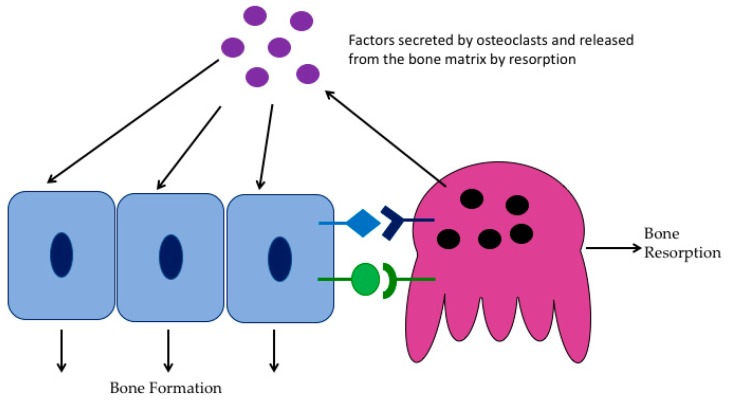
Coupling of bone formation and bone resorption. The cartoon illustrates that osteoblasts express M-CSF and RANKL to stimulate osteoclast differentiation and bone resorption. Osteoclasts also express and release factors that regulate osteoblast differentiation and activity.

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
