# Peer review of "Regulation of Osteoclast Differentiation and Skeletal Maintenance by Histone Deacetylases"

_molecules, 2019, doi:10.3390/molecules24071355_

Round 1

Reviewer 1 Report

In the article “Regulation of Osteoclast Differentiation and Skeletal Maintenance by Histone Deacetylases” the authors provide a review of the field.  The authors initially describe the process of osteoclastogenesis and review the relevant transcription factors that are required for osteoclast differentiation.  A description of Hdac structure and function is provided, and is followed by review of what is know about the function of each Hdac in osteoclasts.  Also included is information on how each Hdac affects the activity of pathways/transcription factors involved in osteoclastogenesis.  The authors end by reviewing the effects of Hdac inhibitors on the skeleton. This is a nice review of the field.  A few comments are noted below.

The authors note in their discussion that more information of the cellular and molecular mechanisms of Hdacs in osteoclasts is needed, as is the function of Hdacs on bone resporption.  This could be expanded upon in the review of the function of each Hdac within the body of the manuscript.  It would be helpful to assess where the field is (e.g., are mouse model lacking, are there not studies on bone resorption, are impacts on TF activity/signaling lacking).

Are there known Hdac mutations in humans that affect bone mass and/or osteoclast differentiation?

Minor points:

Double-check that all instances of special characters (e.g., alpha) are inserted correctly.  Many of them were not.

Could the authors include the personal communication data within this manuscript for publication?

Author Response

1. The authors note in their discussion that more information of the cellular and molecular mechanisms of Hdacs in osteoclasts is needed, as is the function of Hdacs on bone resporption.  This could be expanded upon in the review of the function of each Hdac within the body of the manuscript.  It would be helpful to assess where the field is (e.g., are mouse model lacking, are there not studies on bone resorption, are impacts on TF activity/signaling lacking).

We have added text discussing the areas where there is a lack of knowledge concerning HDAC. These statements were added to the end of each HDAC section describing the gap in the knowledge of the role of that HDAC in regulating osteoclast differentiation as well as some additional text in the introductory sections to the different HDAC classes. 

2.Are there known Hdac mutations in humans that affect bone mass and/or osteoclast differentiation?

We have added a section 4.3 entitled “HDACs and skeletal disease” that discusses the what is known concerning HDAC mutations that affect bone mass.  

Minor points: 

1. Double-check that all instances of special characters (e.g., alpha) are inserted correctly.  Many of them were not.

We have corrected all the special characters to ensure that they are now inserted correctly.  

2.Coud the authors include the personal communication data within this manuscript for publication?

We are currently preparing two manuscripts.  One manuscript will contain the data for the HDAC4 conditional knockout in osteoclasts, and the other manuscript will describe the data for the MEF2A and D conditional knockout mouse model in osteoclasts.  Due to the need to publish these manuscripts, I do not think we can include the personal communication data within this manuscript.   

Reviewer 2 Report

This review on HDACs and osteoclasts has a reasonable review of transcription factors involved in osteoclast differentiation, and lists the HDACs, noting their effects on osteoclasts.  However, the review lacks perspective and the section on HDAC inhibitors is very limited; a recent review (Cantley et al, Bone 95:162) is much more comprehensive.  Most of the individual HDACs seem to promote osteoclastogenesis, and their disruption should improve bone mass, although some have the opposite effect.  However, this is hard to discern from this review.  Figure 1 simply shows when the various family members are expressed but does not indicate any function, and the narrative presentation by class and then number also does not lend itself to providing the reader with an overall view of what HDAC inhibitors (be they broad or specific) would be expected to do.  The "conclusions" section does not even mention HDAC inhibitors or how they might be good or bad for bone. Fig 1 would be improved by color coding the names of the HDACs for their effects (positive, negative, or unknown) on osteoclasts.

One minor point – data from one of the authors should probably be listed as “unpublished observation” rather than “personal communication.”

Author Response

1..The "conclusions" section does not even mention HDAC inhibitors or how they might be good or bad for bone. 

We have added text as well as in the conclusion section of the review discussing whether HDAC inhibitors would be good or bad for bone.  

2.. Fig 1 would be improved by color coding the names of the HDACs for their effects (positive, negative, or unknown) on osteoclasts.

We have modified figure 1 to indicate positive, negative or unknown function of HDACs during osteoclast differentiation. 

3. One minor point – data from one of the authors should probably be listed as “unpublished observation” rather than “personal communication.”

We have changed “unpublished observation” to “personal communication” within the text.  

Reviewer 3 Report

This is an excellent review of the Histone Deacetylases Regulation on Osteoclast Differentiation and Skeletal Maintenance. The introduction and information on the HDACs is well presented and provides a systematic evaluation of each of the HDACS and their current osteoclastogenesis roles. 

As the authors mentioned in the title "skeletal maintenance," I felt that additional information could have been added to the section on HDACS & HDIs about their role in the skeleton. Could they, if any, expand HDACS evidence in specific diseases such as osteoporosis or Paget's disease? Furthermore, are there any evidence of the HDACS or HDIS playing a role in fracture repair or lack of it in non-union fractures. In addition, what are the roles of these factors in bone development, especially in the role of osteoclasts in endochondral bone development, or do any of them play a role in osteocyte - to - osteoblast signaling for osteoclast bone remodeling?

Minor comments 

Page 3 line 94

"d endritic"  - should it be "dendritic"

Page 3 line 101

resorb bone[30].  Space after “bone”

Page 3 line 111

Should it be alpha sign after the C/EBP

Page 3 line 116

"development [38, 39] ." Move period

Page 6 Line 237

“With the increase in osteoclast differentiation there was also an up regulation of

osteoclast genes and an increases osteoclast activity [74].should be ” …and increases in osteoclast activity [74].”

Page 7 line 297

Which osteoclast genes are expressed?

Page 9 line 353

Should be "TNF-α"

Author Response

As the authors mentioned in the title "skeletal maintenance," I felt that additional information could have been added to the section on HDACS & HDIs about their role in the skeleton. 

1. Could they, if any, expand HDACS evidence in specific diseases such as osteoporosis or Paget's disease? 

We have added sections 4.2, 4.3 and 4.4 to discuss what is known concerning HDACs and skeletal maintenance, skeletal disease and bone development. 

2. Furthermore, are there any evidence of the HDACS or HDIS playing a role in fracture repair or lack of it in non-union fractures. 

We have added section 6.1 to discuss what is known concerning HDAC inhibitors and fracture repair.  

3. In addition, what are the roles of these factors in bone development, especially in the role of osteoclasts in endochondral bone development, or do any of them play a role in osteocyte - to - osteoblast signaling for osteoclast bone remodeling?

As stated above for comment #1 we have added section 4.4 which discusses HDACs and their role in skeletal development. 

Minor comments 

We have modified the text to correct all the minor comments listed below by the reviewer. 

Round 2

Reviewer 2 Report

Thank you for your clarifications and addition of color coding to figure.